# Sertoli, Leydig, and Spermatogonial Cells’ Specific Gene and Protein Expressions as Dog Testes Evolve from Immature into Mature States

**DOI:** 10.3390/ani12030271

**Published:** 2022-01-22

**Authors:** Vanmathy R. Kasimanickam, Ramanathan K. Kasimanickam

**Affiliations:** 1Department of Veterinary Clinical Sciences, College of Veterinary Medicine, Washington State University, Pullman, WA 99164, USA; vkasiman@wsu.edu; 2AARVEE Animal Biotech LLC, Corvallis, OR 97333, USA

**Keywords:** canine testis, spermatogonia, Sertoli cells, Leydig cells, mRNA expression

## Abstract

**Simple Summary:**

Sertoli-, Leydig-, and spermatogonial-cells proliferate and differentiate from birth to puberty and then stay stable in adulthood. We hypothesized that expressions of spermatogenesis-associated genes are not enhanced with the increase of these cells’ numbers. To accomplish this postulation, we investigated the abundances of Sertoli cell-specific *FSHR* and *AMH*, Leydig cell-specific *LHR* and *INSL3*, and spermatogonia-specific *THY1* and *CDH1* markers in immature and mature canine testis. Four biological replicates of immature and mature testes were processed, and RT-PCR was performed to elucidate cells’ specific markers. Results showed that the gene expressions of all the studied cells’ specific markers were downregulated in adult testis compared with immature testis. Western blot and immunohistochemistry showed the presence of these proteins in the testis. Protein expressions of these markers were greater in immature compared with mature testis. The results support the postulation that the gene expressions do not directly correlate with the increase of the cell numbers during post-natal development but changes in gene expressions show functional significance.

**Abstract:**

Sertoli, Leydig, and spermatogonial cells proliferate and differentiate from birth to puberty and then stay stable in adulthood. We hypothesized that expressions of spermatogenesis-associated genes are not enhanced with a mere increase of these cells’ numbers. To accept this postulation, we investigated the abundances of Sertoli cell-specific *FSHR* and *AMH*, Leydig cell-specific *LHR* and *INSL3*, and spermatogonia-specific *THY1* and *CDH1* markers in immature and mature canine testis. Four biological replicates of immature and mature testes were processed, and RT-PCR was performed to elucidate the cells’ specific markers. The data were analyzed by ANOVA, using the 2^−∆∆Ct^ method to ascertain differences in mRNA expressions. In addition, Western blot and IHC were performed. Gene expressions of all the studied cells’ specific markers were down-regulated (*p* < 0.05) in adult testis compared with immature testis. Western blot and immunohistochemistry showed the presence of these proteins in the testis. Protein expressions were greater in immature testis compared with mature testis (*p* < 0.05). Despite the obvious expansion of these cells’ numbers from immature to adult testis, the cells’ specific markers were not enriched in mature testis compared with immature dog testis. The results support the postulation that the gene expressions do not directly correlate with the increase of the cell numbers during post-natal development but changes in gene expressions show functional significance.

## 1. Introduction

Tissue-enriched genes are essential for the growth and development of specific cells, tissues, and organs. It has been found that expressions of more than 1000 genes are enriched in the testis. Sertoli, Leydig, and spermatogonial cells actively proliferate and differentiate after birth in all mammalian species, and the enrichment of these cells is considered stable in adulthood. A series of events during fetal and postnatal development of the testis occurs. Proliferation and differentiation of testicular somatic cells and spermatogonia occur during postnatal development [1,2]. These cells’ enrichment has been well studied in mice and rats [1,2,3,4,5]. Spermatogonia increase in number until day 21 [2], and Sertoli and Leydig cells amplify in numbers during the first three to seven weeks [3]. On day 35, complete spermatogenesis is observed in mice. These cell numbers are essentially constant from day 35 in mice [2]. In rats, Sertoli cells cease to proliferate at about 15 days of age [4,5]. Leydig cells are replaced by adult-type cells, and the absolute number of them increases until 60 days of age [4]. On day 13 to 14, the first meiotic cells, preleptotene spermatocytes, are found; the first pachytene spermatocytes are seen between days 19 and 20; the first round of meiotic divisions is completed around day 24; spermatid elongation starts by day 30 to 31, and finally, mature spermatozoa are formed on day 36 [1,5]. Thus, unique processes that occur in germinal cells in the testis, including meiosis, genetic recombination, spermatogenesis, and spermiogenesis, may largely be attributed to several differential gene expressions.

In pigs, Sertoli cells have two distinct proliferation phases postnatally: From birth to 1 month, the cells increase by 6-fold and between 3 and 4 months (just before puberty), the number of cells doubles, whereas Leydig cells proliferate from birth to one month of age and just before puberty. The germ cells proliferate from birth to 4 months with a dramatic increase between 4 and 5 months, and the number stabilizes after 8 months of age [6].

The cell enrichment occurs in dog [7] and boar [6] testis until puberty. Several genes are associated with testicular function and spermatogenesis. Genes such as *PFKB4*, *UROD*, *OCT4*, *PLZF*, *SYCP3*, *SOX2*, *ITG6*, *NEUROG3*, *ID4*, *ITGV*, *THY1*, *GFRA*, *CDH1*, *SMAD5*, *PUM2*, *PRDM14,* and *KIT* were reported as spermatogonia-specific markers [8,9,10,11,12]. Though *OCT4* and *SOX2* are specific to undifferentiated spermatogonia, *OCT4* has been found in A3 spermatogonia [10], and *SOX2* has been found in spermatocytes and spermatids [11,12] in other species. It should be noted that the testis has been identified as the organ in which a large number of tissue-enriched genes are present. However, a large portion of the transcripts related to each stage or cell type in the testes remains unknown. Some studies have claimed that there are no differences, whereas others have claimed differences between immature and mature testes or normal and diseased adult testes [13,14,15,16].

In this study, we intended to investigate Sertoli, Leydig, and spermatogonial cells’ specific markers in immature and adult normal canine testes. We hypothesized that spermatogenesis-associated gene expressions are not enhanced with the mere increase of these cells’ numbers. The objective of the study was to elucidate the cells’ specific markers, Follicle-Stimulating Hormone Receptor (*FSHR*), Anti-Mullerian Hormone (*AMH*) (Sertoli cell-specific), Luteinizing Hormone Receptor (*LHR*), Insulin-Like 3 gene (*INSL3*) (Leydig cell-specific), and Thymocyte antigen 1 gene (*THY1*) and Cadherin-1 gene (*CDH1*) (spermatogonia-specific) in immature and mature canine testes. A further objective was to determine the presence of these proteins in testis using Western blot and immunohistochemistry.

## 2. Materials and Methods

### 2.1. Animals, Testes Collection, and Testicular Tissue Preparation

Testes from healthy dogs undergoing elective castrations were used. The testes from two groups of dogs (immature (2.2 ± 0.13 months; *n* = 4) and mature (11 ±1.0 months; *n* = 4)) were collected. All the surgeries were performed under general anesthesia, and all efforts were made to minimize suffering. There was no evidence for gross lesions on the testes used in this study. Immediately after castration, the testes were maintained in warm Dulbecco’s Phosphate-Buffered Saline (DPBS; Invitrogen, Grand Island, NY, USA). Each testis was decapsulated and epididymis, and superficial blood vessels were removed. Representative samples of testicular parenchyma were cut, snap-frozen in liquid nitrogen (−196 °C) and stored at −80 °C until further use. The testes of four biological replicates of each of immature and mature dog were thawed and processed to elucidate Sertoli, Leydig, and spermatogonial cells’ specific markers, *FSHR*, *AMH*, *LHR*, *INSL3*, *THY1,* and *CDH1*.

The selection of the target genes in this study was based on the microarray-based, high-throughput gene expressions data obtained from the GDS dataset (GDS) of the Gene Expression Omnibus (GEO) repository in the National Center for Biotechnology Information (NCBI) archives (www.ncbi.nlm.nih.gov/geo accessed on: 4 August 2020). For Sertoli, Leydig, and spermatogonia cell-specific markers, one high and one low ranked gene was selected (GDS596).

### 2.2. Real-Time Polymerase Chain Reaction

#### 2.2.1. RNA Isolation

The total RNA isolation was carried out by methods described previously [17]. Briefly, 100 mg of testicular tissue was homogenized in 1 mL TRIzol (Invitrogen, Carlsbad, CA, USA) and incubated for 5 min at room temperature to allow dissociation of nucleoprotein complexes into the TRIzol. After incubation, 0.2 mL chloroform was added to the homogenized samples; the mixture was shaken vigorously and then centrifuged at 12,000× *g* for 15 min at 4 °C. After the centrifugation, the RNA in the aqueous phase was carefully separated, precipitated by adding 0.5 mL of 100% isopropyl alcohol, and centrifuged at 12,000× *g* for 10 min at 4 °C. The RNA pellet was washed with 75% ethanol and dissolved in RNase-free water at 60 °C. A NanoDrop-1000 spectrophotometer (Thermo Scientific, Rockford, IL, USA) was used to measure RNA concentration and determine its quality. The ratio of absorbance at 260/280 ηm was ~2 (1.99–2.05) for all samples. All RNA samples were treated with DNAse I (Invitrogen) to degrade any contaminating DNA and then stored at −80 °C until used.

#### 2.2.2. Complementary DNA (cDNA) Synthesis

Briefly, cDNA was synthesized using the iScript cDNA synthesis kit (Bio-Rad Laboratories Inc., Hercules, CA, USA) [18]. Approximately 500 ƞg of total RNA was added to 20 µL reaction volume and the mixture was incubated at 25 °C for 5 min, 42 °C for 30 min, and at 85 °C for 5 min in a Hybaid PCR Sprint Thermal Cycler (Thermo Scientific). The cDNA (equivalent to 25 ηg/µL RNA concentration) was prepared for each biological replicate and stored at −20 °C until further use.

#### 2.2.3. Real-Time PCR of Cells’ Specific Markers

Specific primer pairs (Table 1) for the cell-specific markers were designed using primer-BLAST (www.ncbi.nlm.nih.gov/tools/primer-blast/ accessed on: 22 September 2020). The primers were first examined with traditional PCR to confirm one amplicon for a primer set. The cDNAs (2μL) were amplified for specific regions of the gene markers, *FSHR*, *AMH*, *LHR*, *INSL3*, *THY1,* and *CDH1,* using Taq-PCR master mix in a Hybaid Thermal Cycler [18]. The reaction mix (20 µL) was prepared using Taq-PCR master mix, primers, cDNA, and nuclease-free water. The DNA amplification steps included: Initial denaturation for 3 min at 94 °C, followed by 30 cycles of denaturation at 94 °C for 1 min, primer annealing for 1 min at 55 °C, and primer extension at 72 °C for 1 min. The final extension step was at 72 °C for 10 min. The DNA template was randomly chosen from the samples, and a reaction was prepared to amplify the expected fragment of a single gene using the singleplex protocol. The immature and mature groups were selected to amplify the fragment of each gene. Negative controls for the template were also included.

#### 2.2.4. Determination of mRNA Expression Using Real-Time PCR

Real-time PCR was performed using Fast SYBR Green Master Mix (Applied Biosystems, Foster City, CA, USA) to study the relative mRNA expression of *FSHR*, *AMH*, *LHR*, *INSL3*, *THY1*, and *CDH1*. Fast SYBR green master-mix (2×) was used to prepare the reaction mix. The final concentration of the primers was 0.3 mM, with 20 µL of three technical replicates used for each sample (2.4 μL of 25 ng/μL RNA equivalent cDNA was present in the total volume of three triplicates). A StepOnePlus instrument (Applied Biosystems) was used, and the enzyme was activated at 95 °C for 20 s, denaturation was done at 95 °C for 3 s, and annealing/extension was carried out at 60 °C for 30 s with forty amplification cycles. Melting curve analysis was also carried out for each target marker. The endogenous control beta-actin [18,19] was used to normalize the Ct values. Fold comparisons were made between the immature and mature testes.

### 2.3. Western Blot Analysis

Western blot analysis was carried out by the methods described previously [20]. Briefly, the testicular tissues from mature and immature dogs were homogenized using ice-cold 1 mL RIPA buffer (Sigma-Aldrich Inc., St. Louis, MO, USA). Protease and phosphatase inhibitor cocktails were added to the RIPA buffer at a volume of 10 µL each to 1 mL of RIPA buffer before homogenization. The homogenized lysate was incubated at 4 °C for 45 min. After incubation, the samples were centrifuged at 12,000× *g* for 20 min, and the supernatants were separated and stored at −20 °C for downstream protein analysis. Protein concentrations of the samples were determined by absorbance at 280 ηm using Nanodrop 1000 spectrophotometer (Thermo Scientific Inc., Waltham, MA, USA). The primary and secondary antibodies for FSHR, AMH, LHR, INSL3, THY1, CDH1, and ACTB used for protein detection are presented in Table 2. Protein lysates (60 μg/lane) were electrophoresed through 12% SDS-PAGE gel (Bio-Rad Laboratories, Philadelphia, PA, USA) and then transferred onto PVDF membrane (Bio-Rad Laboratories, Philadelphia, PA, USA). Non-specific binding was blocked by incubation with 10% goat serum (in which a secondary antibody was produced) in PBS. The membranes were incubated overnight at 4 °C with primary antibodies. After incubation, membranes were washed in a wash buffer containing 2% animal serum and 0.1% detergent and then incubated for 1 h at room temperature in secondary antibodies conjugated with FITC fluorophore. The membranes were then washed and scanned using the Pharos FX Plus system (Bio-Rad Laboratories, Philadelphia, PA, USA) [21,22,23]. The FITC fluorophore was excited at 488 nm and read at the emission wavelength of 530 nm. The immunoblots of β-actin were used for the normalization of the samples [20].

### 2.4. Protein Blot Analysis

Relative quantification was performed for four separately repeated experiments using the Image J software (National Institutes of Health, Bethesda, MD, USA) as described previously [21,22,23]. The relative protein levels were expressed in arbitrary units.

### 2.5. Immunohistochemistry Localization of FSHR, AMH, LHR, INSL3, THY1, and CDH1 on Testis

For the immunohistochemistry (IHC), immature and mature dog testis sections were fixed in 10% neutral phosphate-buffered formalin for 48 h. After washing in phosphate-buffered saline (PBS) and subsequent dehydration in a graded ethanol series, the testis sections were embedded in paraffin. Sections of 4 μm were cut and transferred to Superfrost Plus glass slides. An indirect immunoperoxidase method was performed to detect protein localization in Sertoli, Leydig, and spermatogonial cells. The primary antibodies used were the same as those listed in Section 3.3. The testis sections were deparaffinized in xylol and rehydrated in a graded ethanol series. Antigen retrieval was induced by heat in 10 mM citrate buffer (pH 6.0) for 15 min at 100 °C. Thereafter, the slides were placed in 0.3% hydrogen peroxide in methanol to quench endogenous peroxidases and blocked with 1.5% BSA and 10% goat serum. Incubation with the primary antibodies was carried out overnight in a refrigerated cabinet. Biotinylated goat anti-rabbit IgG or horseradish peroxidase anti-goat IgG (Vector Laboratories, Burlingame, CA, USA) at a dilution of 1:150 was used as the secondary antibody. Signals were enhanced with the avidin/biotinylated peroxidase complex (Vector Laboratories, Burlingame, CA, USA), and the color reactions were achieved using 3,3′-diaminobenzidine (DAB) as chromogen substrate (DAB substrate kit, ab64238, Abcam). Finally, the slides were counterstained with hematoxylin, dehydrated in a graded ethanol series, and covered with coverslips. Negative controls were carried out using rabbit serum instead of the primary antibody.

#### Evaluation of Mean Intensity and Mean Density of the Area Occupied by Chromogenic Labeling for Cell-Specific Proteins in Immature and Mature Testis

To evaluate the mean intensity and mean density of the area occupied by chromogenic labeling for the cell-specific proteins, IHC images were used [20,24]. Image processing and analysis were performed for each testis section sample, and at least 400 cells (Leydig, Sertoli, or spermatogonial cells) of four separate analyses in immature and mature testis were performed [20,24]. The immunohistochemistry images were imported to the software program and were transformed to gray color to determine chromogenic dye localization. Areas (mm^2^) were selected with a similar-sized rectangle using the volume rectangle tool, and the positive staining signal was determined for each protein examined in the randomly selected samples. The background subtraction method was used to obtain the actual target chromogenic labeling in the cells. The same setting was used for all testis section samples. The mean chromogenic intensity (the mean intensity of the pixels inside the volume boundary) and density (the total intensity of all the pixels in the volume divided by the area of the volume) were obtained for each cell type, and the mean (±SD) for a total of 400 cells per sample was calculated.

### 2.6. Stereology

The cell numbers were determined by stereology methods described previously [25,26,27,28]. The testes were fixed for 6 h in Bouin’s solution and then stored in 70% ethanol.

#### 2.6.1. Tissue Preparation and Sampling

Samples were prepared by methods described previously [25,26,27,28]. Initially, each testis was cut to make 8 to 12 slabs of 4-mm-thickness, and every second slab was selected and cut into 6 to 10 bars of 4-mm-thickness. Next, every second bar was selected and cut into 8 to 10 cubes, and finally every fourth cube was selected. Selected tissue blocks were dehydrated in alcohol and embedded in 2-hydroxy-methacrylate. The blocks in methacrylate, each containing 8 to 10 cubes of testicular tissue, were cut into 40 cm-thick sections and stained with Hematoxylin and Eosin to identify Leydig and Sertoli cells. Between 6 and 10 sections were randomly sampled in a systematic manner from each testis.

#### 2.6.2. Cell Identification

Cells, with pale, invaginated, irregular nuclei with a prominent nucleolus in the seminiferous tubules were identified as Sertoli cells [28]. Relatively large, ovoid-shaped cells with an eccentric nucleus containing a prominent nucleolus and peripherally localized chromatin in the interstitium were identified as Leydig cells [28]. Hematoxylin and Eosin were chosen to best identify Leydig and Sertoli cells using strict morphological criteria.

#### 2.6.3. Cell Number Calculation

Total number of cells per testis:$$N=\frac{1}{\mathrm{sf}}*\;\frac{1}{\mathrm{bf}}\;*\;\frac{1}{\mathrm{cf}}\;*\;\frac{1}{\mathrm{ssf}\;}*\;\frac{1}{\mathrm{asf}}\;*\;\frac{1}{\mathrm{hsf}\;}{\;\Sigma Q}^{-}$$
where N = number, sf = slab sampling fraction, bf = bar sampling fraction, cf = cube sampling fraction, ssf = section sampling fraction, asf = area sampling fraction, hsf = height sampling fraction and Σ$Q^{-}$= total cell count [28].

### 2.7. Statistical Analysis

The data were analyzed with a statistical software program (SAS version 9.4 for Windows, SAS Institute, Cary, NC, USA). For all data, normality was tested by PROC UNIVARIATE (Shapiro–Wilk test). Because the raw data were not normally distributed, the transformed (log10) data were analyzed, but the actual non-transformed values were reported. The real-time PCR data were analyzed by one-way ANOVA using 2^−ΔΔCt^ values to ascertain a statistical significance of any differences in the mRNA expressions of *FSHR, AMH, LHR, INSL3, THY1,* and *CDH1* between the mature and immature canine testes. The protein expression data were analyzed by one-way ANOVA. The testis stereology data were analyzed using general linear modeling. When there was a significant overall difference, the significance of the difference between the individual groups was determined by contrast statements. When appropriate, data were logged to avoid the heterogeneity of variance. It was hypothesized that the mean differences in the mRNA expressions will be three-fold between the immature and mature groups. To detect the same differences in the mean mRNA expression, with adequate statistical power (1−β = 0.8) and statistical significance (α = 0.05), the study at least needed a sample size of three dogs per group.

## 3. Results

### 3.1. The mRNA Expression of Cell-Specific Markers

The mRNA expressions of cells’ specific markers, *FSHR* and *AMH* (Sertoli cell-specific), *LHR* and *INSL3* (Leydig cell-specific), and *THY1* and *CDH1* (spermatogonia specific) were less abundant in mature canine testis compared with immature canine testis. Relative mRNA expressions were downregulated to 0.23-fold for *FSHR*, 0.24-fold for *AMH*, 0.31-fold for *LHR*, 0.37-fold for I*NSL3*, 0.12-fold for *THY1,* and 0.02-fold for *CDH1* in the testis of mature dogs compared with immature dogs (Figure 1; *p* < 0.05).

### 3.2. Protein Expressions of Cell-Specific Markers

The presence of immunodetectable FSHR, AMH, LHR, INSL3, THY1, and CDH1 proteins was observed in the Western blots (Figure 2). The immunoblots were also probed for β-actin for normalization of the samples. The antibodies of FSHR, AMH, LHR, INSL3, THY1, and CDH1 were bound to 78 kDa, 65 kDa, 79 kDa, 15 kDa, 18 kDa, 97 kDa, and 44 kDa proteins, respectively (Figure 2). Initially, the quantitative analyses of the relative protein expressions were normalized individually. The protein concentrations (arbitrary units) were greater in the immature testis compared with the mature testis (Table 3; *p* < 0.05). The data were obtained from separate analyses and were expressed as mean ± SD. As an example, immunoblot, optical density plot, and quantitative analyses of the relative protein expressions (arbitrary units) for the FSHR for immature testis and mature testis (normalized) are shown (Appendix A). Additionally, the quantitative analyses of the relative protein expressions were normalized considering all proteins in immature testis and mature testis (Appendix A). The mean protein concentrations (arbitrary units) were greater in the immature testis compared with the mature testis (Appendix A; *p* < 0.05).

### 3.3. Immunolocalization of the Cells-Specific Markers

Image A (Figure 3) shows that FSHR and AMH proteins were localized in Sertoli cells, on Image B (Figure 3), LHR and INSL3 were localized in Leydig cells, and THY1 and CDH1 (Image C, Figure 3) were localized in spermatogonial cells, early-stage cells, and pachytene spermatocytes.

#### The Semi-Quantitative Analysis of the Mean Intensity and Density of the Area Occupied by the Chromogenic Labeling for Cell-Specific Protein Markers in Immature and Mature Dog Testes

In the semi-quantitative analysis of the mean intensity for FSH, AMH, LHR, LNSL3, THY1, and CDH1, the area occupied 51.9, 48.2, 28.1, 34.1, 43.9, and 38.8, respectively, in the immature testis and 28.1, 22.0, 18.9, 19.2, 28.1, and 24.9 in the mature testis, respectively (*p* < 0.05; Figure 4). The densities for FSH, AMH, LHR, LNSL3, THY1, and CDH1 were 421.8, 327.0, 103.0, 212.8, 367.2, and 322.2, respectively, in the immature testis and 245.2, 184.0, 198.8, 132.2, 166.0, and 136.9, respectively, in the mature testis (*p* < 0.05; Figure 4).

### 3.4. The Testis Stereology

The mean total testis weights for the mature and immature dog groups were 15.1 ± 3.4 and 5.7 ± 1.2, respectively (*p* < 0.05). Germ, Leydig, and Sertoli cells were identified based on their morphology (Appendix A). The mean number of germinal, Leydig, and Sertoli cells is shown in Figure 5. The mean mRNA expression of *FSHR, AMH, LHR, INSL3, THY1,* and *CDH1* in immature and mature dog testes is given in Appendix A. In addition, the proportion of mRNA expression of *FSHR, AMH, LHR, INSL3, THY1*, and *CDH1* per testis is given for the immature and mature dogs in Figure 6. The mRNA expression values were normalized as markers per number of cells; (Leydig cells—1056 cells from the immature testis and 176 cells from mature testis; Sertoli cells—3576 cells from the immature testis and 408 cells from the mature testis, and spermatogonial cells—4800 cells for immature testis and 736 cells from mature testis, were assessed; 8E-06% of total cells).

## 4. Discussion

The dog is recognized as a highly predictive model for pre-clinical research. Size, life span, physiology, and genetics make dogs more closely match humans compared with mice [29]. The canine model is much more cost-effective than the non-human primate models [29]. Further, investigations of the genetic basis of disease and new regenerative treatments have frequently taken advantage of canine models [30].

The average ages for the immature and mature dogs used in the current study were 2.2 and 11.0 months, respectively. A study by Lee et al. (2017) showed differentiation of spermatogenic cells from gonocytes to spermatogonia and formation of seminiferous tubules in 1-month-old canine testes, an increase in the number of spermatogonia and Sertoli cells, but no further differentiated stages of germ cells in 2- and 3-month-old canine testes [31]. This data indicated that canine testes remained stable in developmental progression in the prepubertal stage up to 3 months. In 4-month-old canine testes, the seminiferous tubule had expanded, and the germ cells had moved to the basement membrane of the seminiferous tubules. This result indicated that the pubertal stage starts at 4 months in canines. However, fully expanded seminiferous tubules, containing germ cells ranging from spermatogonia to haploid spermatozoa, were identified in the seminiferous tubules from 7-month-old canine testes. These observations exemplified the presence of undifferentiated spermatogonia in prepubertal canine testes, specifically in 1-, 2-, and 3-month-old testes. These results support the categorization of 2-month-olds as sexually immature and 11-month-olds as sexually mature dogs.

Sertoli cells, Leydig cells, and spermatogonia actively proliferate and differentiate after birth in all mammalian species, and the proliferation of these cells is considered stable in adulthood [1,2,6]. We investigated the cells’ specific markers such as *FSHR* and *AMH* (Sertoli cell-specific), *LHR* and *INSL3* (Leydig cell-specific), and *THY1* and *CDH1* (spermatogonia-specific) in the immature and mature canine testes. Gene expressions of these cells’ specific molecular markers were downregulated in the adult canine testis compared with the immature canine testis in the current study. Although there was obvious enrichment of these cells from the immature dog testis to mature dog testis, the expression of these cells’ specific markers was not enriched in the mature testes [32,33,34]. This supports the fact that the genes expressions do not correlate with the mere increase of the cell numbers, but changes in genes expressions warrant functional significance. Therefore, the key consequences of upregulation and downregulation of mRNAs might have been transpired from the event of spermatogenesis (meiosis and differentiation), not from the cells’ enrichments.

In the current study, our analysis revealed that the expression of *AMH* in the Sertoli cells of the immature canine testis was greater than in the mature canine testis. The proteins OD were greater in the immature testis compared with the mature testis. (Appendix A). The *AMH* is not detectable in the normal adult human or animal testis [13,14]. However, the re-expression of the *AMH* has been reported to occur in patients with Sertoli cell-only syndrome [32,34] and Sertoli cell tumors [35,36]. Banco et al. (2012) showed that the Sertoli cells from fetuses and puppies up to 45 days old expressed *AMH*, whereas the SCs from older puppies and adults were negative [15]. All Sertoli cell tumors expressed *AMH* in that study. This re-expression of the *AMH* was also noted in mature, not neoplastic, atrophic testes by Ano et al. 2014 [16]. Interestingly, immunohistochemical analysis study in marmoset monkeys [37] and humans [35,36] revealed that the immune expression of AMH was observed in the Sertoli cells of the newborn and the 8-week-old, but not in the adult. In addition, the expression of FSHR is lower in the adult marmoset testis compared with the 8-week-old [37]. The mRNA expression of the *AMH* and *FSHR* in the newborn, the 8-week-old, and the adult marmoset reflects the increase in the somatic cell numbers during early developmental stages, as well as the declining proportion of the mRNA contributed by the somatic cells following the establishment of spermatogenesis in the adult [32,33,34,35,36]. Considering the above-mentioned finding, we infer that the cell enrichment and expression of cell-specific markers were not correlated during the postnatal developmental progression of the testis.

Leydig cell lineage has five main cell types, specifically the mesenchymal precursor cells, progenitor cells, immature Leydig cells, newly formed adult Leydig cells, and mature Leydig cells [38,39,40]. During the postnatal development, the mesenchymal and Leydig cell numbers increase linearly with a ratio of 1:2 [40]. The onset of precursor cell differentiation into progenitor cells is independent of the LH; however, the LH is essential for the later stages in the Leydig cell lineage to induce cell proliferation and hypertrophy, and to establish the full organelle complement required for the steroidogenic function. Testosterone and estrogen inhibit the onset of the precursor cell differentiation, and these hormones produced by the mature Leydig cells may be of importance in inhibiting further differentiation of the precursor cells to the Leydig cells in the adult testis for maintaining a constant number of Leydig cells [40,41]. Once the progenitor cells are formed, androgens are essential for the progenitor cells to differentiate into the mature adult Leydig cells. Although early studies have suggested that the FSH is required for the differentiation of the Leydig cells, more recent studies have shown that the FSH is not required for this process. Anti-Müllerian hormone has been suggested as a negative regulator in the Leydig cell differentiation, and this concept needs to be further explored to confirm its validity [40]. Insulin-like growth factor I (*IGF-I, INSL3*) induces proliferation of the immature Leydig cells and is associated with the promotion of the maturation of the immature Leydig cells into the mature adult Leydig cells [40].

Zirkin and Ewing (1987) reported that quantitative analyses using light microscopy indicated that the Leydig cell number (per unit volume of testis) was very high in fetal rat testes, significantly low in testes of 2 to 3-day-old rats, and subsequently higher [40,41,42]. When the Leydig cell number was expressed per testis rather than per unit volume of the testis, the testes of the fetal rats and the rats aged 2 to 3 days contained the same number of Leydig cells; after the neonatal period, significant increases in Leydig cell number per testis occurred in concert with the increases in testis weight [40,41,42]. Based on developmental expression patterns, the mutually exclusive expression of INSL3 (a marker of Leydig cell maturity and functionality) defines the mature Leydig cells. Impaired *INSL3* expression has previously been reported in human testicular samples with Leydig cell hyperplasia and adenomas [43]. However, the *INSL3* provides a less ambiguous measure of the Leydig cell functionality, i.e., a combination of Leydig cell differentiation status and the absolute numbers of Leydig cells in the testes [44]. The best-supporting evidence for this is among men who have had one testis removed (monorchid; about half the number of Leydig cells compared with the normal intact men with both testes), the INSL3 levels were significantly lower than in the intact control group, unlike the testosterone levels that did not significantly differ, largely due to the acute compensation by the HPG axis [45]. Second, in the same group of monorchid men, there was a significant negative correlation between the circulating LH and INSL3, reflecting that under conditions where testosterone production is likely to be rate-limiting, the testes with lower Leydig cell functionality will have low INSL3 but induce high LH as part of the HPG homeostatic mechanism to elevate the testosterone. In the current study, the *LHR* and *INSL3* mRNA expressions were in greater abundance in the immature testes compared with the mature canine testes. This result supports the idea that the Leydig cell-specific genes expressions do not correlate with the mere increase of the Leydig cell numbers. Temporal change (upregulation or downregulation) of gene expression from birth to adulthood is associated with the functions of spermatogenesis or other supportive testicular functions. Coding DNA responds to exogenous and endogenous stimuli to become active or inactive, and therefore a gene does not have to be expressed all the time when the proliferation, growth, and differentiation of a single cell population are developmentally progressed.

Several cell surface markers have been reported as spermatogonial stem cell-specific markers in mammals [8,46,47,48,49]. The CDH1, a homophilic cell-cell adhesion molecule spanning the cell membrane is proved to be a type A spermatogonia specific marker [50]. In addition, THY1, a glycosylphosphatidylinositol-anchored glycoprotein of the Ig superfamily is positively expressed in mice spermatogonial stem cells. The *THY1* gene codes thymocyte antigen, which can be used as a marker for a variety of stem cells, and it has been used as an SSC function marker [51,52]. In aquatic species, both *THY1* and *CDH1* genes showed the same expression pattern in different types of testicular germ cells. They were all downregulated in spermatogonia B, spermatocytes, and spermatids. They were more strongly and significantly downregulated in spermatocytes and spermatids, whereas they were highly expressed in spermatogonia and primary spermatocytes [53]. The implication is that these genes are most strongly expressed in spermatogonia A. In the current study, the *THY1* and *CDH1* (spermatogonial cell-specific) mRNA expressions were greater in abundance in the immature testes compared with the mature canine testes. Although the SSC numbers are increased temporally from the immature testis to the mature testis, these germ cell-specific markers’ expressions are downregulated. These findings validate that the change in gene expression depends on the function dictated by the exogenous and endogenous stimuli, but not by the cell proliferation, growth, or differentiation when the organism is developmentally progressed.

The immunohistochemistry of dog testis revealed localization of FSHR and AMH in Sertoli cells, INSL3 and LHR in Leydig cells, and THY1 and CDH1 in spermatogonial cells. The capture of these markers by specific antibodies demonstrates the presence of functional proteins. The present study demonstrated the cellular localization of proteins in the immature and mature dog testes and thus the possibility of the functional significance of these proteins. It should be noted that the protein localization patterns were specific to the cell types. In addition to the localization of proteins, the present study provided information on the semi-quantitative analysis of proteins from the mean fluorescence intensity and fluorescence density analyses. Interestingly, in the current study, the proteins were more abundant in the immature cell types than in the mature cell types. Further, the OD results from the Western blots showed that the proteins were greatly expressed in the immature testis compared with the mature testis. The testis stereology showed that the cell numbers were lower and real-time PCR found that mRNA expression was greater in the immature testis compared with the mature testis. The mean testis weight in the current study was comparable to a previous study [54].

Based on the evidence found in the current and other reported studies, we conclude that the cell enrichment and expression of cell-specific markers were two different phenomena and were not dependent on each other during the post-natal developmental progression of testis in dogs.

## 5. Conclusions

Although there is obvious enrichment of somatic and spermatogonial germ cells from the immature to mature dog testis, the results showed that the expression of the cells’ specific markers was not upregulated from the immature testis to mature testis. The results support the postulation that the gene expressions do not correlate with the increase of the cell numbers, but changes in gene expressions warrant functional significance. We have shown this result, consistently with the series of quantifying/semi-quantifying analyses. Therefore, the key results of upregulation and downregulation of mRNAs might have been transpired from the event of spermatogenesis (meiosis and differentiation), not from the cells’ enrichments.

## Figures and Tables

**Figure 1 animals-12-00271-f001:**
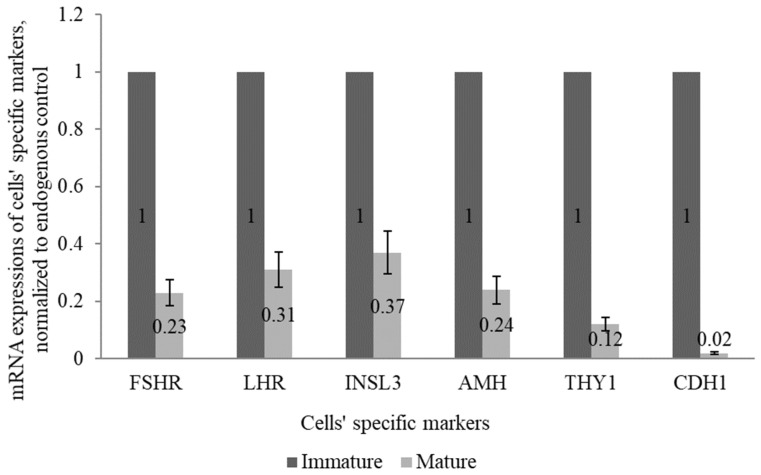
Relative mRNA expression of Sertoli cell-specific Follicle-Stimulating Hormone Receptor (*FSHR*) and Anti-Mullerian Hormone (*AMH*), Leydig cell-specific Luteinizing Hormone Receptor (*LHR*) and Insulin-Like 3 gene (*INSL3*), and spermatogonia-specific Thymocyte antigen 1 gene (*THY1*) and Cadherin-1 gene (*CDH1*). The values between the mature and immature testes are different (*p* < 0.05). The mRNA abundances of all cells’ specific markers were lower in the mature canine testis compared with the immature canine testis (*p* < 0.05).

**Figure 2 animals-12-00271-f002:**
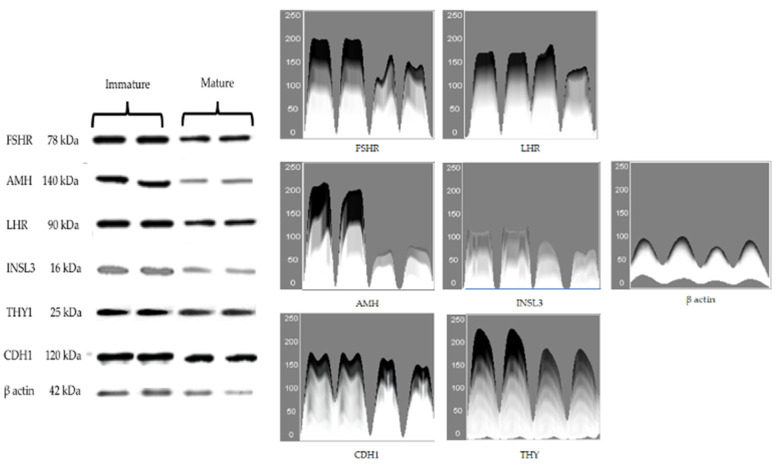
Immunoblots (Western blot) and optical density (surface plots) for FSHR, AMH, LHR, INSL3, THY1, CDH1, and β actin in immature and mature dog testes. FSHR, Follicle Stimulating Hormone Receptor-78 kDa; AMH, Anti-Mullerian Hormone-140 kDa; LHR, Luteinizing Hormone Receptor-90 kDa; INSL3, Insulin-Like 3–16 kDa; THY1, Thymocyte antigen 1–25 kDa; CDH1, Cadherin-1–120 kDa; β actin–42 kDa.

**Figure 3 animals-12-00271-f003:**
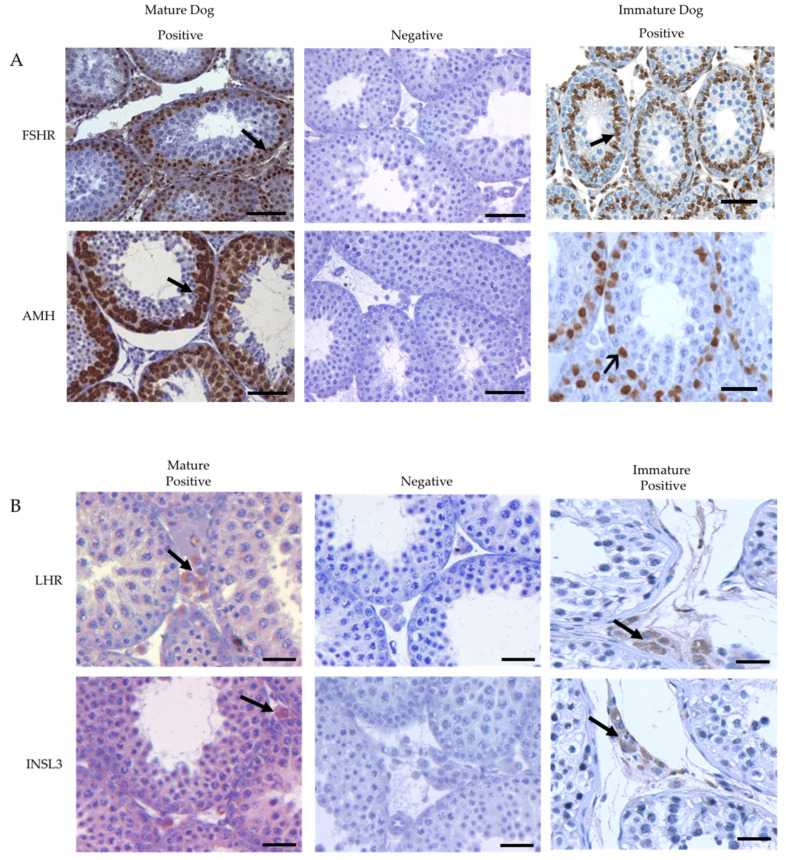
Immuno-localization of testis-specific proteins in mature and immature dog testis. (**A**) The Sertoli cell-specific proteins AMH and FSHR localization in the dog testis (Bar = 50 μm); Note: Only Sertoli cells (arrow) were stained (**B**) Leydig cell-specific proteins LHR and INSL3 localization in the dog testis, Bar = 20 μm); Note: Only Leydig cells (arrow) were stained (**C**) Germ cell-specific proteins THY1 and CDH1 localization in the dog testis, Bar = 50 μm); Note: Only germ cells (arrow) were stained. Refer to Appendix A for cell types.

**Figure 4 animals-12-00271-f004:**
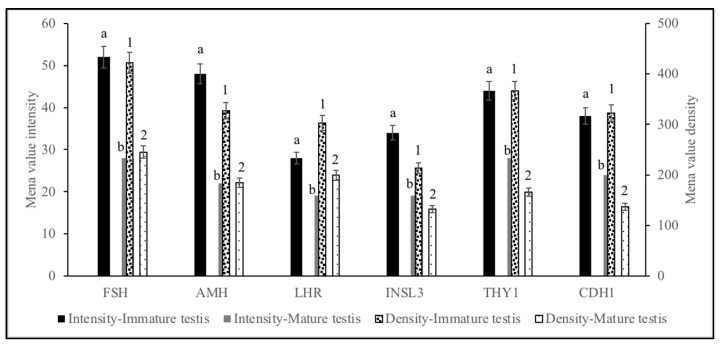
Mean ± SEM (based on four separate analyses) of the intensity and density of the area occupied by the chromogenic labeling for FSH, AMH, LHR, LNSL3, THY1, and CDH1 proteins in immature and mature dog testes. a,b: Bars with different superscripts within the protein, between the mature and immature testes are with different intensities (*p* < 0.05). 1,2 Bars with different superscripts within cell types between the mature and immature testes are with different densities (*p* < 0.05).

**Figure 5 animals-12-00271-f005:**
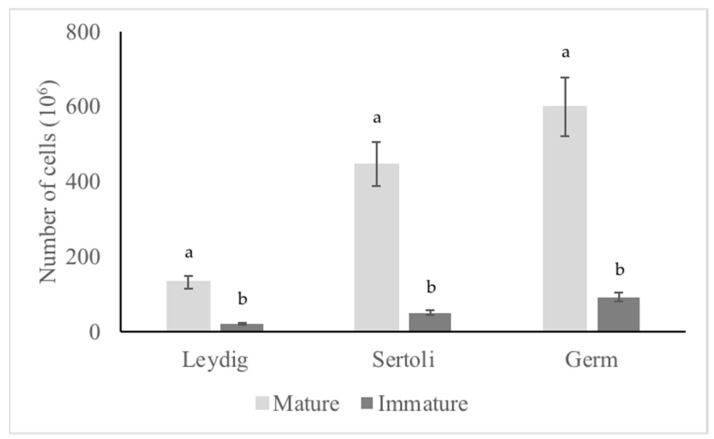
Mean number of germinal, Leydig, and Sertoli cells per testis. a,b: Bars with different superscripts within the cell types, between the mature and immature testes are different (*p* < 0.05).

**Figure 6 animals-12-00271-f006:**
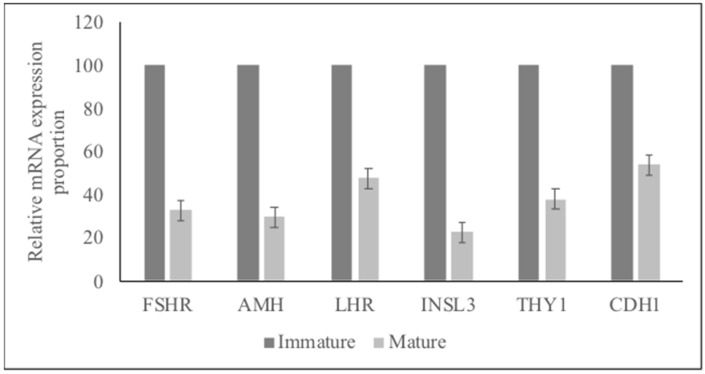
The proportion of mRNA expression *of FSHR, AMH, LHR, INSL3, THY1,* and *CDH1* on testis-basis in mature and immature dogs. Proportions between the mature and immature testes are different (*p* < 0.05).

**Table 1 animals-12-00271-t001:** Forward and reverse primers for canine Follicle-Stimulating Hormone Receptor (*FSHR*), Anti-Mullerian Hormone (*AMH*), Luteinizing Hormone Receptor (*LHR*), Insulin-Like 3 gene (*INSL3*), Thymocyte antigen 1 gene (*THY1*), Cadherin-1 gene (*CDH1*), and β-actin (*ACTB*).

Gene	Primer	Sequence 5′ to 3′	Product Size (Nucleotides)	Accession Number
*FSHR*	Forward Reverse	CGTGTTCTCCAACCTGTCCA TCAGCCATAGTAGAACCTTTTGA	188	XM_003431533.2
*AMH*	Forward Reverse	GGAGGAAGTGACATGGGAGC CAAAGGTTCTGGGTGCCTGG	159	XM_542190.4
*LHR*	Forward Reverse	AACGGTTTCTGCTCACCCAA ATGAGAAAACGGGGTACTGTCA	197	AF389885.1
*INSL3*	Forward Reverse	GACCGTGAGCTGTTGCAGTG CAGTAGTGTGCGGGATTGGT	155	NM_001002962.1
*THY1*	Forward Reverse	GAGCCCAGATCAAGGACTGAG CTGGATGGGCAAGGTGGTAG	179	XM_844606.3
*CDH1*	Forward Reverse	AGGTCTCATCGGGGCTCTG ACACCATCTGTGCCCACTTT	199	XM_536807.4
*ACTB*	Forward Reverse	TCCCTGGAGAAGAGCTACGA CTTCTGCATCCTGTCAGCAA	243	AF021873

**Table 2 animals-12-00271-t002:** Primary and secondary antibodies for canine Follicle-Stimulating Hormone Receptor (FSHR), Anti-Mullerian Hormone (AMH), Luteinizing Hormone Receptor (LHR), Insulin-Like 3 gene (INSL3), Thymocyte antigen 1 gene (THY1), Cadherin-1 gene (CDH1), and β-actin (ACTB) protein detection.

Protein	Primary Antibody	Secondary Antibody
FSHR	Anti-FSH-R antibody (ab137695); rabbit polyclonal	Goat anti-rabbit IgG-FITC (sc-2012)
AMH	Anti-AMH antibody (ab84952: rabbit polyclonal	Goat anti-rabbit IgG-FITC (sc-2012)
LHR	Anti- LHR antibody (ab96603); rabbit polyclonal	Goat anti-rabbit IgG-FITC (sc-2012)
INSL3	Anti-INSL3 antibody (ab199536); recombinant	Goat anti-rabbit IgG-FITC (sc-2012)
THY1	Anti-Thy1 antibody (ab92574); recombinant	Goat anti-rabbit IgG-FITC (sc-2012)
CDH1	Anti-CDH1 antibody (ab226779); rabbit polyclonal	Goat anti-rabbit IgG-FITC (sc-2012)
ACTB	Anti-beta Actin antibody (ab6276); mouse monoclonal	Goat anti-mouse IgG-FITC (sc-2010)

ab: abcam, Waltham, MA, USA; sc: Santa Cruz Biotechnology, Inc. Dallas, TX, USA.

**Table 3 animals-12-00271-t003:** Mean (± SEM) protein expression (optical density, arbitrary units) * in immature and mature canine testes.

Protein	Immature	Mature
FSHR	52.19 ± 3.43 ^a^	31.48 ± 2.27
AMH	49.23 ± 3.34 ^a^	18.42 ± 2.92 ^b^
LHR	31.43 ± 3.84 ^a^	26.38 ± 2.42 ^b^
INSL3	24.19 ± 1.25 ^a^	1912 ± 1.60 ^b^
THY1	51.31 ± 2.24 ^a^	45.43 ± 2.13 ^b^
CDH1	53.44 ± 3.41 ^a^	48.13 ± 1.56 ^b^

^a,b^ Values with different superscripts between the mature and immature testes are different (*p* < 0.05). * Normalized individually (endogenous control, β actin). FSHR, Follicle-Stimulating Hormone Receptor; AMH, Anti-Mullerian Hormone; LHR, Luteinizing Hormone Receptor; INSL3, Insulin-Like 3; THY1, Thymocyte antigen 1; CDH1, Cadherin-1.

## Data Availability

With a reasonable request, the data presented in this study are available from the corresponding author.

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
