# Peer review of "Sertoli, Leydig, and Spermatogonial Cells’ Specific Gene and Protein Expressions as Dog Testes Evolve from Immature into Mature States"

_animals, 2022, doi:10.3390/ani12030271_

Round 1

Reviewer 1 Report

In “Sertoli, Leydig and spermatogonial cells specific gene and protein expressions as dog testes evolve from immature to mature states”, Kasimanickam et. al. investigate the expression of spermatogenesis-associated genes from immature to mature testis. Expression levels of these genes were determined by RT-qPCR, Western blot in testis homogenates and cell localization was determined by immunohistochemistry.

They demonstrated that expression levels of all these markers are diminished in mature testis from dogs compared to immature. Therefore, based on previous studies, they conclude that specific markers expression does not correlate with the increase in cell numbers during post-natal development but might reflects a change regards functionality.

This is the third revision of this manuscript, I consider that this manuscript could be accepted with after clarify two minor comments.

Minor revisions:

1.- In section 2.5.1, authors talk about immunofluorescence quantitation.. I understand that they performed immunocytochemistry (Peroxidase + DAB), so quantitation of the intensity is not possible… They could measure the area of positive or negative signal, but not quantify intensities.

2.- Figure 2 is titled Figure 1.

Author Response

Reviewer 1: 

  1. In section 2.5.1, authors talk about immunofluorescence quantitation. I understand that they performed immunocytochemistry (Peroxidase + DAB), so quantitation of the intensity is not possible… They could measure the area of positive or negative signal, but not quantify intensities.

Authors: The intensity and density calculations relatively measure the positive and negative signals. Therefore, we named as relative or semi-quantification. In other words, it is a relative or comparative measure and we very well addressed this measure in the section 3.3.1.

We also mentioned that the protein expressions were in arbitrary units.

  1. Figure 2 is titled Figure 1.

Authors: Thank you for identifying the error. We corrected accordingly.

Reviewer 2 Report

Dear authors,

This work analyses the expression of genes and proteins of different specific molecules in reproductive male cells of dogs. The manuscript is well written and structured, the introduction provides sufficient background, the research design is appropriate, the results are clearly presented, and the conclusions are supported by the results. The authors should correct the authors form and eliminate the underlined in all the text. Furthermore, minor changes are necessary:

  • Line 161: The authors should add references that support this use, so different studies showed that different housekeeping genes are recommended for different tissues and species.
  • Line 261: before to use a parametric statistical test, the authors should check the normality and homoscedasticity of data. In my experience, gene expression data does not usually have a normal distribution, and a mathematical transformation, usually logarithmic, is usually necessary for its normalization. If so, the authors should specify it.

I hope these suggestions could improve the quality of the manuscript.

Author Response

Reviewer 2:

  • Line 161: The authors should add references that support this use, so different studies showed that different housekeeping genes are recommended for different tissues and species.

Authors: Thank you for the suggestion and we provided and added references

  • Line 261: before to use a parametric statistical test, the authors should check the normality and homoscedasticity of data. In my experience, gene expression data does not usually have a normal distribution, and a mathematical transformation, usually logarithmic, is usually necessary for its normalization. If so, the authors should specify it.

Authors: Thank you for the suggestion and we modified accordingly.

Reviewer 3 Report

In this paper, Kasimanickam and Kasimanickam evaluated the expression of Sertoli-. Leydig- and spermatogonial- cells specific markers in immature and mature canine testis. They found that, generally, these markers were down-regulated in adult compared to immature testis. The results support the authors’ hypothesis that gene expression does not directly correlate with the increase of the cell number during post-natal development.

This is my second review of this paper and, although the authors greatly improved the quality of the MS, further revisions should be made prior its acceptance for publication in Animals:

  • An English editing should be made throughout the text, since many grammar and spell mistakes are present;
  • In sections 2.5.1 and 3.3.1 the authors analyzed the “area occupied by immunofluorescence for cell specific protein…”, but they performed immunohistochemistry, not immunofluorescence; so, how was this data accomplished?
  • In section 3.3, the authors should describe, and then justify, the presence of FSRH and AMH signals also in the germ cells;
  • In Figure 3, arrows and/or other symbols should be added to simplify the reading of the MS; moreover, why did the authors not performed the same experiments on immature testis?

Author Response

Reviewer 3:

  • An English editing should be made throughout the text, since many grammar and spell mistakes are present;

Authors: Thank you for the suggestion and we edited the language including grammar, sentence structure and paragraph flow.

  • In sections 2.5.1 and 3.3.1 the authors analyzed the “area occupied by immunofluorescence for cell specific protein…”, but they performed immunohistochemistry, not immunofluorescence; so, how was this data accomplished?

Authors: Immunofluoroprobe (in this study, secondary antibody is conjugated to the fluoroprobe) is used in immunohistochemistry. Therefore, it is appropriate to say immunofluorescence.

  • In section 3.3, the authors should describe, and then justify, the presence of FSRH and AMH signals also in the germ cells;

Authors: In the Fig C, only THY1 and CDH1(germ cell markers) were probed, and localization image was provided. In Fig A, only FSRH and AMH, (Sertoli cell markers) were probed, and localization image was provided.

  • In Figure 3, arrows and/or other symbols should be added to simplify the reading of the MS; Authors: Description was provided in the foot note.
  • moreover, why did the authors not perform the same experiments on immature testis?

Authors: It was performed (pictures included) and the comparisons were presented.

Round 2

Reviewer 3 Report

The authors addressed almost all the raised issues, improving the quality of the MS. However, their response concerning the immunofluorescence is still not convincing: it is clearly stated, in the text, that they used a secondary antibody conjugated with a peroxidase, the avidin/biotin complex and the DAB as a substrate for the reaction, the fluoroprobe has never been mentioned. Also, Figure 3 clearly shows a immunoistochemistry experiment, so it is hard to imagine how the immunofluorescence intensity has been determined. 

The authors should better clarify this point.

Author Response

Yes, we used DAB in IHC for chromogenic labeling.

We used wrong terminology. "immunofluorescence" was replaced with  "chromogenic labeling". Thanks. 

Round 3

Reviewer 3 Report

The authors followed all my suggestion improving their paper. It is suitable for publication in Animals in this form.

This manuscript is a resubmission of an earlier submission. The following is a list of the peer review reports and author responses from that submission.

Round 1

Reviewer 1 Report

In “Cells enrichment does not influence gene and protein expression of Sertoli, Leydig and spermatogonial cells specific markers in canine testis”, Kasimanickam et. al. investigate if expression of spermatogenesis-associated genes from immature to mature testis. Expression levels of these genes were determined by RT-qPCR, Western blot in testis homogenates and cell localization was determined by immunohistochemistry.

They found that, expression levels of all these markers are diminished in mature testis from dogs compared to immature after normalization to b-actin. Therefore, based on previous studies, they conclude that specific markers expression does not correlate with the increase in cell numbers during post-natal development but might reflects a change regards functionality.

This is the second revision of this manuscript, Even when testis stereology has improved this manuscript, I consider that this manuscript still needs more analysis and modifications.

Major revisions:

1.- Reading the coverletter, I agree with other reviewer, it is evident that not all genes increase their expression levels as cells numbers increase. So the title should be replaced by something similar to “Expression of gene and protein markers as dog testes evolve from immature and mature states”. The authors maintained almost the same title, I think it should be modified.

2.- As I stated in my previous revision, the analysis in expression levels was performed in testis homogenates and normalization to b-actin, so I wondered if expression changes could be due to changes not in the total number of cells per testy but changes in the proportion of different cell types. To overcome this issue, the authors have improved the manuscript providing a key determination, testis stereology. Thus, in Figure 3 they show that the number of Leydig, Sertoli and Germ cells increase in mature testis compared to immature. Thus, decrease in the molecular targets does not reflects a change in the number of each cell type.

However, the description of the results is not clear in the Results section. I would normalize expression levels of Sertoli markers to the number of Sertoli cells, the levels of Leydig markers to the number of Leydig cells, etc. Reading the manuscript, I couldn´t understand how was determined the proportion of mRNA expression showed in figure 4. This should be described in Results or Materials and Methods sections.

This section of results should be described in more detail and also I think that expression levels normalized to the number of each specific cell type should be determined.

3.- In my first revision of this manuscript I asked to the authors if expression levels in Table 2 are normalized to b-actin. Also I suggest to show the density plots in Supplemental. Authors moved the hole Figure to supplemental. I think that protein expression is very important and should be shown as a Figure rather than supplemental. More important, the quality of the blot´s pictures are worse compared to the first manuscript, the contrast of the pictures seems to be very high, so I can not appreciate the technical quality of the western blot and the background in each band.

Also, looking at the b-actin representative figure, still seems that quantitation of the western blots presented in Table 2 is not normalized to b-actin.

I also noted in the first manuscript that CDH1 membrane was cutted, showing one peace with immature and other with mature testis (these peaces present different the background). In the second manuscript, the blot seems to be the same. The contrast is very high to determine if the same problem is happening but it appear to be the same image just with different contrast. I think authors should replace this image by other representative blot.

Minor comments:

1.- In line 279, “Tetis Stereology” should be corrected to “Testis Stereology”.

Reviewer 2 Report

Dear authors,

This study analyses the differences in expression of genes and proteins between mature and immature dog testis. The manuscript is well structured and written, the introduction provides sufficient background, the results are clearly presented, and the conclusions are supported by the results. However, major revisions are necessary in research design, description of methods and minor revisions in introduction:

  • Line 50: eliminate point before reference 3.
  • Lines 67-69: some of these genes, as OCT4 and SOX2 are related to undifferentiation cells, but in spermatogonia. The authors should change this sentence, so some of these genes are not spermatogonia specific markers.
  • Line 160: Why have the authors used beta-actin as housekeeping? Many studies indicate that the choice of the best housekeeping depends on the type of fabric, species, etc., so the authors should justify their choice.
  • Line 246-247: the authors must verify the normality and homoscedasticity of the data. In my experience, expression data usually have non-normal distributions, and non-parametric statistical methods should be applied, or a mathematical transformation (usually logarithmic) should be performed before carrying out a parametric analysis such as ANOVA.
  • Line 248: the names of genes must always be written in italics.

I hope that these recommendations improve the quality of the manuscript.

Reviewer 3 Report

In this paper, the authors intended to verify whether the expression of Sertoli-, Leydig- and spermatogonia specific genes was correlated to the increase of these cells’ number in immature and mature dog testis. They analyzed, via RT-PCR, western blot and immunohistochemistry analyses, the levels of Sertoli cell specific FSHR and AMH, Leydig cell specific LHR and INSL3, and spermatogonia specific THY1 and CDH1 markers.

They found that gene expressions of all cells' specific markers studied were down-regulated in adult compared with immature testis

The paper is interesting; however, it requires an extensive revision prior its publication in Animals:

  • English should be edited throughout the text, were spell mistakes (line 227), repetition (line 402) and other errors (IMH in line 30, the reference in line 426) are present
  • In line 33, genes should be replaced with protein, since western blot and IHC both detect proteins
  • In the section 2.3, the dilutions of the used antibodies are totally missing; in the section 2.5, since the antibodies codes are the same as in section 2.3, they can be omitted
  • Why did the authors perform the IHC just on mature testis?
  • Sections 3.3 (Immunolocalization of cells specific markers) and 3.3 (that should be 3.4, testis stereology) should be completely rewritten, since the authors did not give any information of the results
  • The figure concerning the western blot result should also be changed, since, conventionally, bands and graphs are shown; moreover, in Fig. S1, the actin bands should be approximately the same in all the samples (in mature testis they are both weaker than in immature)
  • In the discussion section, it would be more appropriate if the authors were not referring to others’ papers as “results”
  • In lines 442 - 444, the authors assessed that “the immunohistochemistry of dog testis revealed localization of FSHR and AMH in Sertoli and spermatogonial cells, INSL3 and LHR in Leydig and Sertoli cells and THY1 and CDH1 in Sertoli, Leydig and spermatogonial cells”, leading to the conclusion that the used antibodies are not specific
  • Since the authors analyzed just two “specific” markers per cell-type, their conclusion that “the genes expressions do not correlate with the mere increase of the cell numbers” is too much general, and should be reconsidered; moreover, additional interpretation on why the analyzed genes follows such trend should be given.